# Parthenolide as Cooperating Agent for Anti-Cancer Treatment of Various Malignancies

**DOI:** 10.3390/ph13080194

**Published:** 2020-08-14

**Authors:** Malgorzata Sztiller-Sikorska, Malgorzata Czyz

**Affiliations:** Department of Molecular Biology of Cancer, Medical University of Lodz, 6/8 Mazowiecka Street, 92-215 Lodz, Poland; malgorzata.sztiller-sikorska@umed.lodz.pl

**Keywords:** parthenolide, herbal medicines, cancer treatment, NF-κB, multimodal therapy, synergistic effect, cancer stem-like cells

## Abstract

Primary and acquired resistance of cancer to therapy is often associated with activation of nuclear factor kappa B (NF-κB). Parthenolide (PN) has been shown to inhibit NF-κB signaling and other pro-survival signaling pathways, induce apoptosis and reduce a subpopulation of cancer stem-like cells in several cancers. Multimodal therapies that include PN or its derivatives seem to be promising approaches enhancing sensitivity of cancer cells to therapy and diminishing development of resistance. A number of studies have demonstrated that several drugs with various targets and mechanisms of action can cooperate with PN to eliminate cancer cells or inhibit their proliferation. This review summarizes the current state of knowledge on PN activity and its potential utility as complementary therapy against different cancers.

## 1. Introduction

Parthenolide (PN) is a sesquiterpene lactone derived from leaves of the medicinal plant feverfew (*Tanacetum parthenium*). It contains an α-methylene-γ-lactone ring and epoxide group, which are able to interact with nucleophilic sites of biological molecules (Figure 1) [1]. For centuries, PN has been used as a herbal medicine with regard for its anti-inflammatory and anti-migraine properties, as reviewed extensively [2,3,4]. Recently, the anticancer potential of PN has attracted great attention. PN, and its 1000-fold more soluble in water synthetic derivative, dimethylamino-parthenolide (DMAPT; Figure 1), were subjects of many preclinical in vitro and in vivo studies, performed on cancer cells from hematological malignancies and solid tumors [5,6,7,8,9,10,11,12,13,14,15,16,17,18,19,20,21,22,23,24,25,26,27,28,29,30,31,32,33,34,35,36,37,38,39,40,41,42,43,44,45,46,47,48,49,50].

A distinctive feature of PN is its ability to induce cell death in cancer cells while sparing normal ones [12,22]. Moreover, PN significantly reduces the viability of both the bulk population of cancer cells and the cancer stem-like cell subpopulation [14,17,22,39,40,41,42,43,44,45,46]. PN has been reported to trigger a robust apoptosis in primitive CD34^+^ cell population from acute myelogenous leukemia (AML) specimens while sparing normal hematopoietic cells [22] and can induce cell death in primitive chronic myelogenous leukemia (CML) cells through reactive oxygen species [46]. PN has been shown to reduce the frequency of ABCB5^+^ melanoma cells in melanospheres [14], induce oxidative stress, mitochondrial dysfunction and necrosis of breast cancer stem-like cells [17] and target prostate cancer CD44^+^ cells [44].

## 2. Mechanisms of PN Action

The mechanisms of PN-induced anticancer activity are complex. PN predominantly acts via the inhibition of nuclear transcription factor-kappa B (NF-κB) (Figure 2, Section 1) [1,13,23,51]. NF-κB is constitutively active in many types of cancer and controls expression of genes important for tumor growth, angiogenesis, and metastasis [52,53,54,55]. Elevated activity of NF-κB may have essential effects on sensitivity of cancer cells to therapy and has been attributed to development of drug resistance [56,57,58]. PN inhibits the activity of the upstream kinases of IκB complex (IKK) precluding degradation of the of two NF-κB regulatory proteins, IκB-α and IκB-β [51], and/or it directly alkylates the NF-κB/p65 subunit at Cys38 and Cys120 residues (Figure 2, Section 1) [59].

Suppressed NF-κB activation is correlated with sustained c-Jun N-terminal kinase (JNK) activation followed by tumor necrosis factor α (TNF-α)-mediated cell death of cancer cells [60]. PN-induced JNK activation can be also independent of NF-κB inhibition [61]. PN can modify the activity of p53 along two mechanisms, both involving ataxia telangiectasia mutated (ATM) serine/threonine kinase (Figure 2, Section 2). It can occur either by promoting the ubiquitination and degradation of the mouse double minute 2 homolog (MDM2), a p53 negative regulator, or by depletion of histone deacetylase 1 (HDAC1) through proteasomal degradation (Figure 2, Section 2) [20,21]. Inhibition of HDAC1 activity by PN might also reduce the level of M isoform of microphthalmia-associated transcription factor (MITF-M) [23]. Furthermore, PN also is an effective inhibitor of signal transducer and activator of transcription (STAT) proteins. By blocking STAT3 phosphorylation on Tyr705, PN prevents STAT3 dimerization, its nuclear translocation, and STAT3-dependent gene expression (Figure 2, Section 3) [6,29,33]. Inversely, PN does not interfere with the phosphorylation of STAT6 but inhibits its DNA-binding activity [62]. PN suppresses the activity of ubiquitin-specific peptidase 7 (USP7) and WNT/β-catenin signaling [63]. Oxidative stress may also contribute to the therapeutic effects of PN. PN-induced apoptosis is preceded by the decrease of the levels of intracellular thiols, including both free glutathione (GSH) and protein thiols, and generation of reactive oxygen species (ROS) (Figure 2, Section 4) [7,38,64]. Besides induction of apoptosis, PN can also generate ROS-mediated autophagy [19,37]. α-methylene-β-lactone moiety of PN that is highly reactive with intracellular thiols has been found to be indispensable for the inhibitory effect of PN on tubulin carboxypeptidase [65,66], which might reverse the accumulation of abnormal detyrosinated tubulin observed in cancer cells. PN has been reported to be an effective DNA hypomethylating agent acting as inhibitor of DNA methyltransferase 1 (DNMT1) expression and activity [67].

However, PN also induces undesirable, cellular protective responses that likely reduce its overall cytotoxicity and may limit its clinical application, such as activation of nuclear factor erythroid 2-related factor 2/antioxidant response elements (Nrf2/ARE) pathway. Induction of phase II detoxification/antioxidant enzymes and their transcriptional targets e.g., heme oxygenase 1 results in increased resistance to oxidative damage [39,68]. In melanoma cells, PN increases the level of phosphorylated extracellular signal-regulated kinase (ERK1/2) [23,69], which is associated with upregulation of connective tissue growth factor (CTGF) expression [69], involved in melanoma cell migration and invasion [70]. This unfavorable effect could be blocked by trametinib, an inhibitor of mitogen-activated protein kinase 1/2 (MEK1/2) [69].

Impact of PN on important signaling pathways in cancer development provided a basis for the studies combining approved drugs and PN in various types of cancers. This review presents a wide range of approved drugs and experimental compounds that have been reported to cooperate with PN.

## 3. Parthenolide in Combination with Anticancer Agents

### 3.1. Parthenolide Combined with Tubulin-Directed Agents 

Taxane diterpenoid anticancer agents such as docetaxel (Taxotere) and paclitaxel (Taxol) have shown a significant potency against many tumor types including metastatic breast cancer, advanced ovarian cancer, non-small-cell lung cancer (NSCLC), prostate cancer and Kaposi’s sarcoma [71,72,73]. Paclitaxel and docetaxel arrest cells at the G2/M phase of cell cycle and induce cell death by stabilizing microtubules and interfering with microtubule disassembly during cell division [74,75]. Major weakness of using taxanes in cancer therapy is the development of drug resistance, which has been associated with the activation of NF-κB [76,77,78].

Several studies involving multiple experimental approaches and diverse cancers suggest that PN might be used in combination with taxanes to enhance anticancer effects. Effective dose of paclitaxel necessary to induce cytotoxic effect in human NSCLC cell lines was markedly reduced by PN [79]. PN precluded NF-κB nuclear translocation and activation as well as BCL-XL up-regulation caused by paclitaxel in human lung cancer cell lines sensitive (A549, NCI-H446) and resistant (A549-T24) to paclitaxel [80]. Most likely, this effect was derived from reduced BCL-XL induction via PN-mediated modulation of the NF-κB/I-κB kinase (IKK) signal cascade (Figure 2, Section 1) [79]. Inhibition of NF-κB by PN promoted cytochrome c release, activation of caspase-3 and -9 and inhibition of transcription of several anti-apoptotic proteins, including c-IAP1 and BCL-XL [79,80]. Co-treatment of A549 cells with PN and paclitaxel resulted in activation of the intrinsic pathway of apoptosis in contrast to mitochondria-independent apoptosis induced by paclitaxel used alone [80]. Likewise, in A549 tumor mice xenografts PN increased the efficacy of paclitaxel [81]. Combination of drugs caused a greater inhibition of the tumor growth compared with paclitaxel alone, while PN alone was unable to suppress tumor development [81]. Combination therapy was also accompanied with decreased angiogenesis and significant increase in the life span of xenograft mice and was well-tolerated [81]. Additional benefits were gained from co-encapsulation of PN with paclitaxel in mixed micelles, which increased the anticancer activity against paclitaxel sensitive and resistant NSCLC cell lines and caused significantly higher cell death than drugs used in solution [82].

Activated NF-κB also protected breast cancer cells against paclitaxel. PN increased the sensitivity of these cells to paclitaxel and enhanced paclitaxel-induced apoptosis by inhibition of constitutive NF-κB DNA binding activity (Figure 2, Section 1). In MDA-MB-231 cells, it was associated with reduced manganese superoxide dismutase (Mn-SOD) expression [76]. In gastric cancer cells the combined treatment of paclitaxel and sub-lethal dose of PN significantly increased apoptosis compared with the cells treated with paclitaxel alone [34]. The PN/paclitaxel combination significantly elongated survival of mice in the peritoneal dissemination model. PN also enhanced chemosensitivity to paclitaxel [34].

Docetaxel in combination with PN induced apoptosis in breast cancer cells, reduced the colony formation capacity and expression of a prometastatic gene, CXCL8, and an antiapoptotic gene, GADD45β1 [83]. Mice bearing MDA-MB-231 cell xenograft treated with this drug combination showed remarkably enhanced survival associated with reduced lung metastases in comparison with animals treated with each drug alone or untreated animals [83]. PN also increased the in vivo efficacy of docetaxel in a xenograft of androgen-independent prostate cancer mouse model. Docetaxel in combination with PN decreased tumor growth more efficiently compared with control xenografts and xenografts treated either with docetaxel or with parthenolide alone [32]. On the other hand, in androgen-dependent LNCaP prostate cancer cells exerting low basal NF-κB activity, PN blocked docetaxel- and 2-methoxyestradiol-mediated apoptosis [84]. Thus, combination of antimitotic drugs with NF-κB inhibitors can have antagonistic effects.

Vinorelbine, a semi-synthetic vinca alkaloid, induces cytotoxicity by disruption of microtubule assembly and is approved for the treatment of advanced breast cancer and NSCLC [85,86]. Both PN and vinorelbine inhibited proliferation in human breast cancer cell lines MCF7 and MDA-MB-231, but vinorelbine was less effective in side-population (SP) cells that represent cancer stem-like cells than in non-SP cells [87]. By using vinorelbine in combination with PN, the proliferation of SP cells was significantly inhibited. In the MCF7 xenografts in mice, stealth liposomal vinorelbine together with stealth liposomal PN produced a strong inhibitory effect in breast tumors [87].

### 3.2. Parthenolide Combined with TRAIL

Tumor necrosis factor-related apoptosis-inducing ligand (TRAIL) is a cytokine considered as a promising anticancer agent as binding of TRAIL to its receptors, death receptor 4 (DR4) and death receptor 5 (DR5) initiates apoptosis in various cancer cell types without affecting normal cells [88,89]. However, various cancers develop resistance toward TRAIL [89,90]. Co-treatment with PN is considered as capable to overcome resistance to TRAIL [6,25,61,91].

TRAIL used in combination with PN significantly inhibited cell proliferation and induced apoptosis in TRAIL-resistant and TRAIL-sensitive human colorectal cancer cells (CRC) [25,91]. Synergistic effect was associated with upregulation of DR5 protein and its cell surface expression, deregulation of BCL-2 family members, increase in p53 level, release of cytochrome c to cytosol and activation of caspases [25,91]. Similar effects were observed in human hepatocellular carcinoma (HCC) cells lines [6]. The expression of death receptors DR4 and DR5 either at the protein or mRNA levels as well as their surface localization were increased in PN-treated HCC cell lines. Probably, this is a consequence of an inhibitory effect exerted by PN on the activation of JAK proteins resulting in decreased level of phosphorylated STAT proteins (Figure 2, Section 3) [6]. Co-treatment of HCC cell lines with PN and TRAIL induced apoptosis along extrinsic pathway, and this was associated with the activation of both caspases-8 and -3, whereas a mitochondrial pathway was not involved [6]. In contrast, PN-induced TRAIL sensitization did not alter the expression of death receptor in breast cancer cells, but activation of JNK was necessary for this process [61]. TRAIL used in combination with PN induced apoptosis in TRAIL-resistant MDA-MB-231 cells [61]. PN induced TRAIL-dependent cleavage of Bid and XIAP degradation that was associated with increased caspase-3 activity and poly (ADP-ribose) polymerase (PARP) cleavage, whereas activity of caspase-8 was not affected [61]. 

### 3.3. Parthenolide Combined with Anti-Inflammatory Drugs

NF-κB and cyclooxygenase (COX) are activated or overexpressed in several cancers, including pancreatic cancer [92,93]. Nonsteroidal anti-inflammatory drug (NSAID) sulindac, a COX inhibitor, suppressed the growth of pancreatic cancer cells [94]. 

Sulindac in combination with PN lowered the threshold for apoptosis and synergistically inhibited proliferation of pancreatic carcinoma cells. The PN/sulindac combination cooperatively targeted the NF-κB pathway, which was observed as increased level of IκBα protein, and decreased NF-κB transcriptional activity compared with either agent used alone [94]. Moreover, sulindac combined with DMAPT exerting higher bioavailability than PN, might mediate their antitumor effects by altering cyclin D1 levels, also in a xenograft model of human pancreatic cancer [95]. DMAPT combined with sulindac delayed or prevented progression of premalignant pancreatic lesions in the developmental mouse model of pancreatic cancer, in which tumors are formed de novo. The percentage of normal pancreatic ducts was significantly increased by the DMAPT/sulindac combinations compared with placebo [96]. 

Celecoxib, a COX-2 inhibitor/NSAID used in combination with DMAPT prevented from adjacent organs invasion and metastasis in an in vivo model of pancreatic cancer. It was related to decreased prostaglandin E2 and prostaglandin E2 metabolite levels and reduced NF-κB activity [97]. PN decreased the effective dose of NS398, a COX-2 inhibitor that was used for growth inhibition of human hepatocellular carcinoma cells [98]. The combination of PN and NS398 more efficiently induced G1 cell cycle arrest, decreased cyclin D1 levels and inhibited NF-κB DNA-binding and transcriptional activities than the single agents. Drugs used in combination inhibited phosphorylation of the NF-κB-inhibitory protein IκBα and increased total IκBα levels [98]. 

Balsalazide is a colon-specific prodrug of 5-aminosalicylate used in the treatment of inflammatory bowel disease [99]. Inhibition of NF-κB activity is one of the several mechanisms of reduction of inflammatory responses by balsalazide [100,101]. PN and balsalazide produced synergistic anti-proliferative effects and induced apoptosis through the mitochondrial pathway in human colorectal carcinoma cell line and colitis-associated colon cancers (CAC) [100,101]. The combination of balsalazide and PN markedly suppressed the nuclear translocation of NF-κB, p65 and IκBα phosphorylation, NF-κB binding to DNA and expression of NF-κB targets [100,101]. Combined treatment of a CAC murine model resulted in a regain of body weight and suppressed carcinogenesis [100,101].

### 3.4. Parthenolide Combined with Hormonal Agents

Tamoxifen, a nonsteroidal selective estrogen receptor antagonist, is commonly used in the treatment of recurrent estrogen receptor-α-positive (ER+) breast carcinoma [102]. A major problem of this therapy is a primary resistance and acquired resistance induced by drug [102]. Inhibition of NF-κB may also be useful for the treatment of ER+ breast cancers that have acquired resistance to antiestrogen therapy. MCF7/HER2 and BT-474 breast cancer cell lines overexpressed the oncogenic receptor tyrosine kinase, HER2/ErbB2, which activates NF-κB and diminishes the responsiveness to tamoxifen [103]. Tamoxifen used in combination with PN suppressed activity of NF-κB- and activator protein 1 (AP-1) and increased nuclear receptor co-repressor (NCoR) binding to the promoter of ER encoding gene [104,105].

NF-κB inhibition by PN (Figure 2, Section 1) synergistically restored sensitivity of resistant breast cancer cells to 4-hydroxytamoxifen (4HT), and further sensitized drug-naïve cells to 4 HT [106]. In primary ER+ solid tumors, tamoxifen with PN generated significant tumor responses that were estrogen-independent [107].

Steroidal antiestrogen fulvestrant (Faslodex) stimulates degradation of the ER, prevents receptor dimerization, and inhibits estrogen-dependent gene transcription [108]. Resistance to this drug is also connected with enhanced expression and activity of NF-κB [109].

MCF7/LCC9, antiestrogen-resistant breast cancer cell line was insensitive to fulvestrant, while PN efficiently inhibited proliferation of these cells. The combination of these drugs caused a 4-fold synergistic reduction of proliferation [110]. PN also restored sensitivity to anti-androgen therapy in a xenograft hormone refractory prostate cancer model (androgen-independent prostate cancer line CWR22Rv1 xenograft model). In mice treated with the combination of anti-androgen bicalutamide and PN, tumor volume was smaller than in controls and in mice treated with single agents and had a slower rate of proliferation [32].

Dehydroepiandrosterone (DHEA) is a naturally occurring adrenal steroid, a precursor of sex steroids such as androgen and estrogen. Among a variety of additional, beneficial effects of DHEA is inhibition of the NF-κB signaling pathway [111]. Combined treatment of DHEA and PN efficiently inhibited growth and function of AtT20 corticotroph tumor cells both in vitro and in vivo. In vitro, DHEA and PN had inhibitory effects on pro-opiomelanocortin and NF-κB-dependent gene expression. Combined administration of DHEA and PN in mice with xenografts of AtT20 cells significantly reduced tumor growth and survivin expression, decreased corticosterone levels in plasma and eliminated the malnutrition induced by tumor growth [112].

### 3.5. Parthenolide Combined with Anthracyclines

Doxorubicin (Dox), isolated from *Streptomyces peucetius*, is a potent anticancer agent and a treatment of first choice for many cancers, but is not highly effective against melanoma in vitro and in vivo [113,114]. However, its efficacy was substantially increased when it was combined with PN. Dox caused pro-survival effects including NF-κB activation and stimulation of the ATP-binding cassette sub-family B member-5 (ABCB5) transporter expression. PN markedly lowered Dox-induced NF-κB activity and reduced expression of matrix metalloproteinase-9 (MMP9) [36]. PN also synergized with Dox against malignant glioblastoma cell lines [115].

A similar effect was observed in human colon cancer cells after treatment with PN and disaccharide analog of Dox, sabarubicin (MEN-10755) [116]. As Dox caused an early pro-survival NF-κB activation, sensitivity of HCT-116 cells to drug-induced apoptosis was low. Inhibition of NF-κB by PN (Figure 2, Section 1) increased sabarubicin-induced cell death [116].

In human breast cancer cells, PN prevented the development of resistance to mitoxantrone and doxorubicin by inhibition of overexpression of proteins involved in the protection against oxidative stress, e.g., manganese superoxide dismutase (MnSOD) and other proteins that support the survival of cancer cells and favor drug resistance, including heat-shock proteins 70 (HSP70), BCL-2 and P-glycoprotein [117].

### 3.6. Parthenolide Combined with Alkylating Agents

Dacarbazine (DTIC), an alkylating agent, is used for the treatment of metastatic melanoma [118]. PN in combination with DTIC synergistically reduced the number of viable melanoma cells and abolished the baseline and DTIC-induced vascular endothelial growth factor (VEGF) secretion [26], however, the mechanisms of the synergy between these drugs remain unclear. 

Temozolomide (TMZ) and radiotherapy are used for the treatment of glioblastoma multiforme (GBM), the most malignant brain tumor in adults [119]. While orally administrated temozolomide provides survival benefits for GBM patients, the overall clinical prognosis is unsatisfactory due to intrinsic or acquired resistance to TMZ [120]. It has been shown that high expression of the DNA repair enzyme O6-methylguanine-DNA methyltransferase (MGMT) is responsible for resistance to TMZ in GBM cells [121]. Anti-apoptotic activity of NF-κB has also been reported as being associated with resistance to TMZ [122], and NF-κB was found to play a major role in regulation of MGMT [123]. Inhibition of NF-κB activity by PN (Figure 2, Section 1) reduced the expression of MGMT and restored sensitivity to temozolomide in LN18 and T98G glioma cell lines [124]. PN in combination with temozolomide more efficiently inhibited tumor growth and prolonged median survival in human T98G xenograft-bearing severe combined immunodeficiency (SCID) mice than these drugs used as single agents [124].

Cisplatin is a chemotherapeutic drug used since late 1970s to treat numerous cancers [125,126]. Cisplatin and other platinum-containing drugs crosslink with the purine bases on the DNA causing DNA damage and affecting DNA repair mechanisms, which results in apoptosis of cancer cells. PN markedly enhanced the sensitivity of human lung cancer cells to oxaliplatin, an analog of cisplatin, by inhibiting NF-κB activation (Figure 2, Section 1) and inducing apoptosis [127]. PN synergistically increased the anticancer effect of cisplatin in both apoptotic and sub-apoptotic concentrations in GBM rat and human cell lines, overcoming cisplatin resistance in GBM rat cells, probably through NF-κB inhibition [115]. PN could also reverse cisplatin resistance of human gastric carcinoma cells [27]. In the cisplatin resistant human gastric cancer cell line, PN and cisplatin demonstrated time- and concentration-dependent cytostatic effects, and a synergy between PN and cisplatin was demonstrated [27]. Cisplatin acted synergistically in vitro and in vivo also with PN-based compound ACT001 (dimethylamino-micheliolide) [128]. In human GBM cell line, combination of cisplatin and ACT001 increased the apoptosis and reduced cell migration and invasion more efficiently than each drug alone. In the mouse xenograft model, inhibition of tumor growth was significantly enhanced by combined treatment in comparison to single agent treatment. Synergy between ACT001 and cisplatin was related to the inhibition of the plasminogen activator inhibitor-1 (PAI-1)/PI3K/protein kinase B (AKT) pathway [128]. 

PN has been reported to act as an epigenetic drug inhibiting both HDAC1 [20] and DNMT1 [67]. In combination with 5-aza-2′-deoxycytidine, an inhibitor of DNMT1, PN synergistically reduced growth and induced apoptosis of medulloblastoma-derived cells [129]. 

### 3.7. Parthenolide Combined with Antimetabolites

Ganciclovir (a synthetic analogue of 2′-deoxy-guanosine), an antiviral medication could be use in Burkitt lymphoma (BL) therapy, because Epstein-Barr virus (EBV) is present in cancer cells of almost all patients with endemic variant of BL. Elevated NF-κB levels in EBV-positive BL cells inhibit lytic replication cycle of the latent EBV [130]. Treatment with PN inhibited proliferation of Raji cell line and led to apoptosis via the mitochondrial pathway. Moreover, inhibition of RelA/p65 activity by PN induced EBV lytic replication. Ganciclovir significantly augmented cytotoxic effect of PN in Raji cells [28].

Thus, 5-fluorouracil (5-FU) is an important element of chemotherapy in many types of cancer [131]. It blocks DNA synthesis by inhibition of thymidylate synthase. PN can overcome 5-FU resistance in human colorectal cancer cells [132]. Combination of these drugs synergistically induced apoptosis through the mitochondrial pathway in human colorectal cancer cells. Moreover, intra-peritoneal injection of PN and 5-FU resulted in inhibition of tumor growth and induction of apoptosis in the mouse CRC xenograft model [132]. PN reversed the 5-FU resistance also in hepatic carcinoma resistant cells [133]. Most recently, 23 conjugates of parthenolide and 5-FU were evaluated in hepatocellular carcinoma cell line [134], and one of these conjugates exerted improved anticancer activity when compared with PN alone. 

Gemcitabine, a synthetic pyrimidine nucleoside prodrug, is used in the treatment of a wide variety of tumors e.g., lung, breast, ovarian and pancreatic cancer [135]. It has been shown that gemcitabine can induce NF-κB activity in human pancreatic cancer cell lines, which can diminish its clinical efficacy [136]. DMAPT, when used in combination with gemcitabine, revoked NF-κB induction in pancreatic cancer cells and intensified gemcitabine’s anti-proliferative effect [136]. In a mouse model of pancreatic cancer, DMAPT and gemcitabine co-treatment increased the percentage of normal ducts and decreased the percentage of intraepithelial neoplasia-2 lesions [96].

### 3.8. Parthenolide Combined with Histone Deacetylase Inhibitors

The antitumor activity of histone deacetylase inhibitors (HDACIs) has been related to their ability to generate accumulation of hyperacetylated histones and alter gene transcription [137]. HDACIs can downregulate the expression of oncoproteins, induce differentiation, cell cycle arrest and apoptosis, and some of them are under clinical development [137,138,139]. As several HDACIs have been shown to induce NF-κB activity leading to cytoprotective effects, NF-κB inhibitors such as PN can potentiate apoptosis induced by HDACIs [140,141,142]. Moreover, it has been reported that PN also specifically depletes HDAC1 protein (Figure 1, Section 2) [20].

Valproic acid, an antiepileptic drug with well documented pharmacokinetics and toxicity profiles exerted an HDAC-inhibitory activity in cancer cells like neuroblastoma or teratocarcinoma [143]. As a single agent, it had slight growth-inhibitory activity in cultured cancer cells but induced the NF-κB transcriptional activity [142]. PN markedly synergized with valproic acid to induce apoptosis in esophageal cancer cells, NSCLC cells and malignant pleural mesothelioma cells [142].

Suberoylanilide hydroxamic acid (SAHA, vorinostat), an HDACI with a high therapeutic potential has been shown to act synergistically with PN in a triple-negative breast cancer cell line (Figure 3) [144]. SAHA/PN combination decreased the intracellular GSH level and mitochondrial membrane potential, released cytochrome c, activated caspase-3 and apoptosis more potently than each drug alone [144]. These two drugs used in combination induced epigenetic changes such as hyperacetylation of histones H3 and H4 as the result of SAHA-induced inhibition of HDAC and down-regulation of DNMT1 induced by PN. They caused depletion of GSH, increased expression of the tumor suppressor genes p21 and p27 and reduced the Bcl-2 and p65 level. Moreover, combined treatment reduced the prosurvival effects observed in single-drug treatments: induction of autophagy by SAHA and induction of AKT/mammalian target of rapamycin (mTOR)/nuclear factor erythroid 2-related factor 2 (NRF2) pathway by PN [144].

PN cooperated synergistically with SAHA and another pan-histone deacetylase inhibitor LBH589 also in acute myeloid leukemia (AML) cells [11]. PN inhibited HDACI-induced IKK and RelA phosphorylation and stimulated MKK7-dependent activation of stress-activated protein kinase/c-Jun N-terminal kinase (SAPK/JNK) pathway [11]. It leads to apoptosis in multiple AML cell lines, transduced hematopoietic cells with leukemia-initiating cell characteristics and primary AML blasts, while sparing normal hematopoietic progenitors. A series of parthenolide-SAHA hybrids were synthesized and tested for their anti-cancer properties [145]. The most active hybrid showed several times higher activity against AML cells than PN or SAHA alone and induced apoptosis through mitochondria pathway [145].

In some cases, co-treatment with PN and HDACI could have negative effects. It has been demonstrated that trichostatin A, the reversible HDAC inhibitor, promoted acetylation of NF-κB/p65 subunit [146]. Co-treatment of human colon adenoma cells with trichostatin A and PN significantly inhibited the trichostatin A-induced cellular levels of acetyl NF-κB/p65 and decreased NF-κB/p65 binding to the promoter of putative tumor suppressor gene, stanniocalcin-1 (STC1) [147]. 

### 3.9. Parthenolide Combined with mTOR Inhibitors

Inhibitors of the phosphoinositide 3-kinases (PI3K)/mTOR pathway have been shown to synergize with PN in selective eradication of primary human AML cells [39,148]. The combination of PN and PI3K inhibitor wortmannin or the mTOR inhibitor, rapamycin (sirolimus) and rapamycin analog, temsirolimus exhibited synergistic toxicity against AML stem cells. These effects were manifested both in vitro with patient specimens and in vivo using murine xenotransplantation models of AML [39,148].

Ciclopirox, an antifungal agent, used for the treatment of mycoses, has shown an antitumor activity attributed also to inhibition of mTOR [149]. Similarly, to other mTOR inhibitors, ciclopirox significantly enhanced the ability of PN to target acute myeloid leukemia by inhibiting the PN-induced activation of mTOR. Ciclopirox and PN acted synergistically to eradicate bulk and leukemia stem cells [149].

### 3.10. Parthenolide Combined with Retinoids

All-trans retinoic acid (Tretinoin, ATRA) induces remission in a majority of acute promyelocytic leukemia patients [150]. ATRA forced promyelocytic leukemia cells to differentiate and inhibited their proliferation [151]. PN increased ATRA-induced HL-60 cell differentiation into a granulocytic lineage involving activation of PKC, ERK2, JNK and PI3-K and downregulation of asparagine synthetase expression [152,153]. Close association between enhancement of cell differentiation and inhibition of NF-κB DNA-binding activity by PN was shown [152].

Fenretinide, a synthetic retinoid with antineoplastic activities, induced the activation of NF-κB during apoptosis [154]. Sublethal dose of PN effectively enhanced the fenretinide-induced apoptosis in hepatoma cells [154].

Vildagliptin, an inhibitor of dipeptidyl peptidase-4 is used clinically as an anti-hyperglycemic drug in the treatment of type 2 diabetes mellitus [155]. High throughput combination drug screen revealed that vildagliptin synergistically enhanced anti-leukemic activity of PN in primary AML samples including the stem cell fraction [41]. Vildagliptin used in combination with PN decreased the viability and clonogenic growth of AML cells, whereas spared normal human peripheral blood stem cells. Synergy between drugs was not related to vildagliptin’s primary target dipeptidyl peptidase 4. Enhancement of parthenolide’s activity was mediated by inhibition of dipeptidyl peptidase-8 and dipeptidyl peptidase-9 by vildagliptin [41].

### 3.11. Parthenolide Combined with Inducers of Reactive Oxygen Species (ROS)

Arsenic trioxide (ATO), a well-known environmental toxicant, has been approved for the treatment of certain types of leukemia [156]. PN increased cytotoxicity of arsenic trioxide in cell lines representing various hematological malignancies (EL-4, Jurkat and HL-60 cells) [9]. Addition of buthionine sulfoximine, an irreversible inhibitor of γ-glutamylcysteine synthetase that reduces intracellular GSH levels, increased efficacy of treatment. The glutathione content and intracellular ATP levels were reduced, and induction of oxidative stress was observed. Lymphocytes from healthy donors were mostly resistant to this treatment [9]. Most recently, significant increase in the cytotoxicity of arsenic trioxide by PN was confirmed in adult T-cell leukemia/lymphoma cell lines [157]. Combination therapy with arsenic trioxide and PN generated reactive oxygen species, inhibited growth, and induced apoptosis via the mitochondrial pathway also in human pancreatic carcinoma cell lines [158]. In pancreatic carcinoma xenografts in nude mice PN/ATO treatment inhibited tumor growth more efficiently than either drug alone [158]. 

Okadaic acid is a lipophilic polyether produced by several marine dinoflagellates. It acts by inhibiting serine/threonine protein phosphatases [159]. Okadaic acid/PN treatment effectively induced apoptosis, increased ROS levels and reduced GSH level in human retinoblastoma Y79 cells [160]. These two compounds at subtoxic doses acted together to induce a tumor suppressor PTEN, p53 activation and decreasing phospho-AKT and phospho-MDM2 levels [160].

### 3.12. Parthenolide Combined with Other Drugs

Abnormal activation of FMS-like tyrosine kinase 3 (FLT3) is observed in AML patients [161]. FLT3 selective inhibitor SC-203048 in combination with PN synergistically inhibited AML growth and increased cell apoptosis [162]. 

Heat-shock proteins (HSP) are often overexpressed in human solid tumors and hematological malignancies [163]. HSP90 inhibitor, geldanamycin induced changes in the apoptosis-related protein levels, loss of the mitochondrial transmembrane potential, formation of reactive oxygen species, nuclear damage, and cell death in the human epithelial ovarian carcinoma cell lines [164]. PN enhanced geldanamycin-induced apoptosis by increasing activation of the caspase-8- and Bid-dependent pathway and the mitochondria-mediated apoptosis leading to the activation of caspase-9 and caspase-3. The stimulatory effect of PN on the formation of reactive oxygen species (Figure 2, Section 4) was probably crucial for this combined action [164].

The main mechanism of PN action in cholangiocarcinoma (CC) was also probably associated with oxidative stress, caused by GSH depletion and ROS generation [24]. Simultaneously, during induction of apoptosis by PN in CC cells, heme oxygenase-1 was highly expressed, which could cause cell resistance to chemo-oxidative stress [165]. Protein kinase C-alpha inhibitor Ro317549, which blocked heme oxygenase-1 expression through downregulation and inhibition of nuclear translocation of Nrf2, enhanced PN-mediated apoptosis. The combination of PN and Ro317549 also efficiently inhibited tumor growth in a subcutaneous tumor model [165].

Actinomycin-D is a cyclic polypeptide antibiotic that binds to DNA and prevents elongation of RNA chain by inhibiting RNA polymerase II [166]. A phase 2 clinical trial has shown that actinomycin-D could be an effective drug in the pancreatic cancer treatment; however, it has substantial dose-limiting toxic side effects [167]. Actinomycin-D combined with DMAPT synergistically enhanced cell death of pancreatic cancer cells and may provide an effective treatment at lower doses [168].

Most cancer cells remain in a highly proliferating state because they lost the ability to mature into non-replicating adult cells. Certain cancers may be treated with agents that induce terminal differentiation [169]. Thus, 1α,25-dihdroxyvitamin D3 has been shown to potently initiate differentiation of promyelocytic leukemia cell into a monocytic lineage. Pretreatment of these cells with PN strongly potentiated the 1α,25-dihdroxyvitamin D3-induced cell differentiation via the inhibition of NF-κB activity (Figure 2, Section 1), although PN alone did not induce cell differentiation [170].

Lactacystin, an inhibitor of proteasome activity in combination with PN led to activation of the caspase cascade and synergistically increased the apoptotic fraction of the p53-deficient, drug-resistant mouse leukemia L1210 cell (Y8) [171]. 

Nowadays, feverfew supplements containing at least 0.2% PN are broadly available and can be purchased in the form of capsule, tablet, or liquid extract. It has been demonstrated that PN together with other constituents of feverfew, flavonoids (apigenin and luteolin) in feverfew extract might have moderate to weak synergistic effects on the inhibition of growth of human breast cancer cells and human cervical cancer cells [172].

## 4. Parthenolide in Combination with Radio- and Thermotherapy

### 4.1. Parthenolide Combined with Radiotherapy

Radiotherapy is currently used to treat localized tumors such as cancers of the skin, brain, breast, and prostate, but it is delimited by side effects and late complications resulting from high-dose radiation and development of resistance [173]. The acquired radio-resistance involves several mechanisms, including the activation of NF-κB. Ionizing radiation has been reported to activate NF-κB in both in vitro and in vivo models [174,175]. Furthermore, constitutive activity of NF-κB may be responsible for the intrinsic radioresistance of cancer cells [30]. Treatment with PN is considered as the possibility to decrease therapeutic doses of radiation and reduces radiation toxicity to surrounding cells.

Inhibition of constitutive and radiation-induced NF-κB activity by PN (Figure 2, Section 1) sensitized prostate cancer cells to X-ray, inhibited expression of superoxide dismutase 2 (SOD2) gene and split-dose repair and activated the PI3K-AKT pathway [176,177]. PN also mediated intense oxidative stress in these cells by enhancing ROS generation and attenuating antioxidant protection by activation of the reduced form of nicotinamide adenine dinucleotide phosphate (NADPH) oxidase and downregulation of MnSOD and catalase. When PN was used with radiation, further increase of ROS levels was observed in prostate cancer cells [35]. Contrarily, PN increased GSH level and reduced radiation-induced oxidative stress in normal prostate epithelial cells PrEC [35].

Treatment with DMAPT decreased proliferation and enhanced radiation-induced cell death in NSCLC cells [178,179] and human prostate cancer cells [180]. DMAPT increased the efficiency of single and fractionated X-ray treatment through inhibition of a constitutive and radiation-induced NF-κB activity and inhibition of DNA double-strand break repair [178,179,180]. NF-κB inhibition reduced homologous recombination, nonhomologous end joining and downregulated ionizing radiation-induced DNA repair biomarkers [178]. DMAPT significantly decreased tumor growth in A549 lung cancer tumor xenografts and PC-3 prostate tumor xenografts in nude mice, compared with xenografts treated with either DMAPT or radiation alone [179,180]. 

Treatment of whole-body X-irradiation of Transgenic Adenocarcinoma of the Mouse Prostate (TRAMP) mice with a normal immune system with PN or DMAPT prior to combination of low and high doses, protected normal tissues from a radiation-induced apoptosis [31]. Both drugs induced radiosensitivity in TRAMP prostate tissue, particularly in areas with higher grade prostatic intraepithelial neoplasia lesions [31].

Osteosarcoma is a primary malignant bone tumor, commonly demonstrates significant radioresistance and high metastatic potential [181]. In osteosarcoma, human and murine cell line, resistance to radiotherapy has been associated with NF-κB activity [182,183]. Pretreatment with PN synergistically suppressed cell proliferation and potentiated apoptosis induced by ionizing radiation in osteosarcoma cells. Moreover, both the overall population of cancer cells and the CD133+ stem-like cell subpopulation were affected. This effect was connected to the ability of PN to induce oxidative stress [182]. PN with irradiation treatment inhibited osteosarcoma tumor growth in mice more effective compared to PN or irradiation alone [183]. PN enhanced X-ray sensitivity of radiation resistant CGL1 (HeLa hybrid) cells [30]. Treatment with PN induced stabilization of p53, blocked cell cycle and caused inhibition of split-dose repair [30].

### 4.2. Parthenolide Combined with Thermotherapy

Hyperthermia has shown promising therapeutic outcomes as an element of combined therapy to treat human cancers [184,185,186]. Elevated body temperature causes immune system activation and can alter the physiology of cancer cells by activating heat shock proteins, induction protein unfolding, enhancing oxygenation of the tumor microenvironment and blood flow to the tumor [184,185,186], but cancer cells can develop thermo-resistance or thermotolerance similar to radiotherapy and chemotherapy [185].

PN and elevated temperature used in combination caused significant sensitization of the human lung adenocarcinoma A549 cells to apoptosis. Synergistic thermo-enhancement effects on induction of apoptosis and cell cycle arrest in G2/M were due to the direct suppression of NF-κB activity (Figure 2, Section 1) [187,188]. PN was also an effective thermo-sensitizing agent for human prostate cancer therapy [189]. Treatment of human prostate cancer androgen-independent cell lines with PN prior to hyperthermia significantly increased apoptosis and caused G2/M cell cycle arrest with the participation of the mitogen-activated protein kinase (MAPK) cascade [189].

## 5. Conclusions and Perspectives

PN, when used alone, has been identified in preclinical studies as an effective agent against many types of cancer and has been proposed to increase the sensitivity of cancerous cells to chemotherapy and radiotherapy (Table 1). Multimodal therapies that include PN seem to be an interesting conception to simultaneously eliminate different cancer subpopulations, including cancer stem-like cells and provide more complete eradication of the malignancy. PN and other naturally derived agents (e.g., curcumin, betulin, withanolides, lactoferrin) when administered with the standard platinum and etoposide treatments led to a complete regression of advanced small lung cell carcinoma and the patient remains in full remission more than 7 years, while the median survival with the standard treatments alone is 9–10 months from the time of diagnosis [190]. While a wide array of biological activity and low toxicity make PN a promising drug with multi-pharmacological potential, its bioavailability is unsatisfactory, which limits its clinical application in anticancer therapy. A lot of effort has been made to overcome its instability in both acidic and basic conditions and improve its solubility and bioavailability [191,192,193,194,195,196,197,198,199]. Several new water-soluble PN derivatives have been designed, synthesized and their activity has been evaluated in vitro and in vivo [43,47,193,194,195,196,197,198,199,200,201,202]. Better solubility and higher anticancer activity have been recently obtained with PN derivatives prepared through the aza-Michael addition of nitrogen-containing anticancer drugs such as cytarabine and melphalan [200]. PN and some of PN-based anti-cancer candidates have been clinically tested (Table 2). 

In a phase I dose escalation trial, PN given daily as an oral Feverfew tablet was well tolerated without dose-limiting toxicity but did not provide detectable plasma concentrations [18]. ACT001 (dimethylamino-micheliolide (DMAMCL) fumarate salt) developed by Accendatech Co was certified by Food and Drug Administration (FDA) and European Medicines Agency (EMA) as an orphan drug for the treatment of gliomas, and it is used in the clinical trials in Australia [201] and China. Besides new derivatives of PN with better bioavailability, alternative delivering systems are in the focus of current research. Encapsulation into the micelles, drug-loaded nano-vectors or nanographene delivery [16,203,204,205,206] might allow to exploit the full spectrum of the anticancer activities of PN and may be crucial for the development of new combination therapies.

Finally, it is worth mentioning that a potent anti-inflammatory activity of PN also contributes to the scientific interest in this compound [2,207,208]. Most recently, PN has been suggested for clinical evaluation as a part of anti-cytokine storm therapy in patients with pneumonia caused by severe acute respiratory syndrome coronavirus 2 (SARS-CoV-2) [209].

## Figures and Tables

**Figure 1 pharmaceuticals-13-00194-f001:**
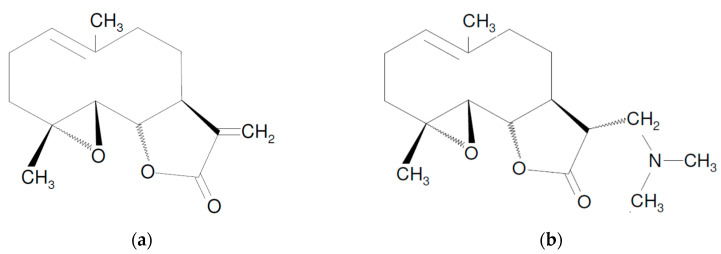
(**a**) Chemical structure of parthenolide (PN), a sesquiterpene lactone from Tanacetum parthenium, and (**b**) Chemical structure of dimethylamino-parthenolide (DMAPT), a synthetic, water-soluble derivative of PN.

**Figure 2 pharmaceuticals-13-00194-f002:**
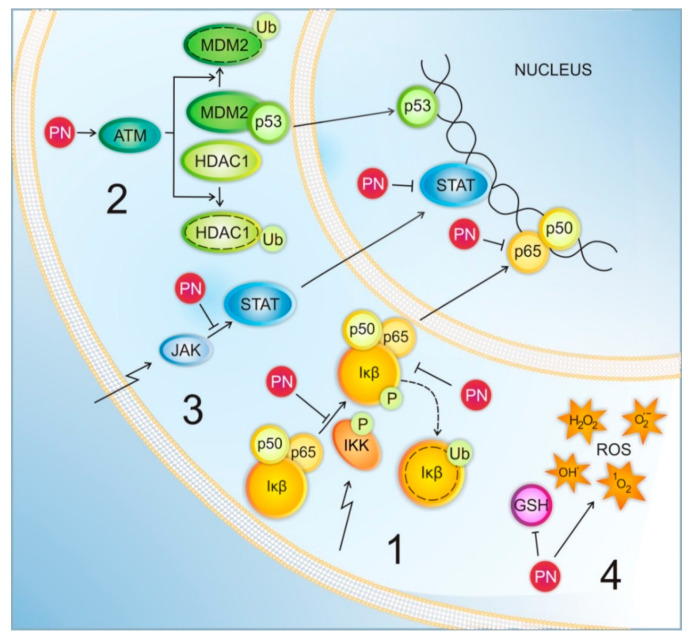
Selected mechanisms of anticancer activity of parthenolide (PN). (**1**) Inhibition of nuclear factor kappa B (NF-κB) signaling either through interaction with IκB kinase (IKK) or directly with p65 subunit results in the downregulation of antiapoptotic gene transcription; (**2**) Degradation of the mouse double minute 2 homolog (MDM2) and depletion of histone deacetylase 1 (HDAC1) regulate activity of p53 and increase cancer cells sensitivity to therapy; (**3**) Inhibition of signal transducer and activator of transcription protein 3 (STAT3) phosphorylation prevents its dimerization, nuclear translocation and STAT3-dependent gene expression; (**4**) Reduction of cellular level of glutathione (GSH) and reactive oxygen species (ROS) accumulation induce oxidative stress.

**Figure 3 pharmaceuticals-13-00194-f003:**
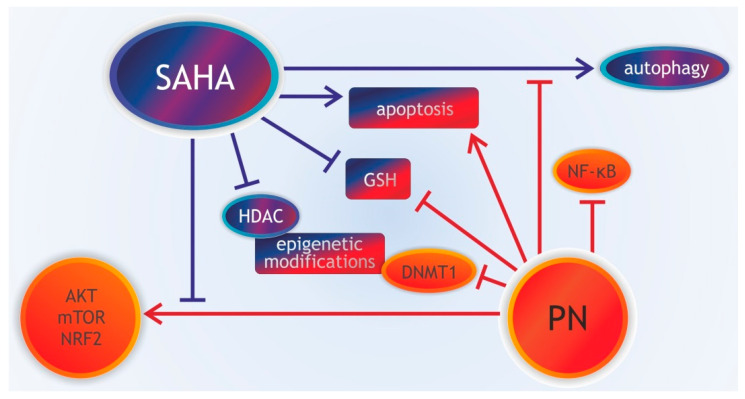
Schema summarizing synergistic effects of suberoylanilide hydroxamic acid (SAHA) and parthenolide PN in triple-negative breast cancer cells prepared based on Carlisi et al., [144]. SAHA (histone deacetylase 1 inhibitor, HDACIs) and PN used in combination lead to several epigenetic modifications as SAHA induces hyperacetylation of histones H3 and H4 and PN downregulates DNA methyltransferase 1 (DNMT1). These modifications, together with reduced glutathione (GSH) depletion induced by combined treatment with SAHA and PN, trigger apoptosis. While PN restrains pro-survival autophagy that is induced by SAHA, SAHA prevents cytoprotective effects of PN on AKT/mammalian target of rapamycin (mTOR)/nuclear factor erythroid 2-related factor 2 (NRF2) pathway. SAHA does not influence the inhibitory effect of PN on nuclear factor kappa B (NF-κB) activity.

**Table 1 pharmaceuticals-13-00194-t001:** Chemotherapeutics investigated in combination with parthenolide (PN) and its derivatives in different types of cancer.

Co-Treatment with PN	Main Mechanism of Action	In Vitro Cell Line (PN Concentration)	In Vivo Model (PN Dose)	References
Leukemias				
1α,25-dihdroxyvitamin D3	differentiation inducer	human HL-60 (0.625–10 μM)	−	[170]
ATO (arsenic trioxide)	reactive oxygen species inducer	human Jurkat, human HL-60, human K-562 (10 μM)	−	[9]
ATRA (all-trans retinoic acid)	differentiation inducer	human HL-60 (0.25–10 μM)	−	[152,153]
ciclopirox	mTOR inhibitor translation initiation factor eIF5A inhibitor	human Kasumi-1, human primary AML specimens (2.5–10 μM)	−	[149]
SC-203048	FLT3 selective inhibitor	−	human THP-1 in athymic BALB/c nude mice (10 μg/kg 7x each 2nd day)	[162]
lactacystin	proteasome inhibitor	murine L1210 (Y8) (5 μM)	−	[171]
rapamycin/ temsirolimus	mTOR inhibitors	human primary AML specimens (5 μM)	human primary AML cells in NOD/SCID mice (100 mg DMAPT/kg 3x daily)	[39]
SAHA (suberoylanilide hydroxamic acid) LBH589	pan-histone deacetylase inhibitor	human U-937, human HL-60, human NB4, human MV-4-11, human MOLM-13(3–8 μM)	−	[11]
vildagliptin	dipeptidyl peptidase IV inhibitor	human TEX, human OCI-AML2 (2.5–5 μM)	−	[41]
wortmannin	PI3Ks inhibitor	human primary AML specimens (2.5–10 μM)	−	[39]
Breast Cancers				
4HT (4-hydroxytamoxifen)	selective estrogen receptor modulator	human MCF7/RR, human MCF7/LCC1, human MCF7/LCC9 (0.5 μM)	−	[106]
docetaxel	microtubules stabilizer	human HBL-100, human MDA-MB-231 (0.5–5 μM)	human MDA-MB-231 in female nude mice (40 mg/kg daily)	[83]
doxorubicin mitoxantrone	topoisomerase II inhibitor	human MDA-MB-231 (1–10 μM)	−	[117]
faslodex (fulvestrant)	estrogen receptor down-regulator	human MCF7/LCC9 (0.6 μM)	−	[110]
paclitaxel (Taxol)	microtubules stabilizer	human MDA-MB-231 (0.1–5 μM)	−	[76]
SAHA	pan-histone deacetylase inhibitor	human MDA-MB-231 (2.5–25 μM)	−	[144]
tamoxifen	selective estrogen receptor modulator	human MCF7, human MCF7/HER2, human BT-474 (1–50 μM)	−	[103,104,105,107]
TRAIL (tumor necrosis factor-related apoptosis-inducing ligand)	apoptosis inducer	human MDA-MB-231 (1–5 μM)	−	[61]
vinorelbine	microtubule assembly blocker mitosis inhibitor	human MCF7, human MDA-MB-231 (0.5–10 μM)	human MCF7 in female BALB/c nude mice (10 mg/kg 5th and 8th day after inoculation)	[87]
Colorectal Cancers				
5-fluorouracil	thymidylate synthase inhibitor	human SW620 (10 μM)	human SW620 in nude mice (2.5 mg/kg 3x weekly)	[132]
balsalazide	anti-inflammatory agent	human HCT116, human SW480, human HT-29 (5–10 μM)	azoxymethane-induced CAC in BALB/c female mice (2 mg/kg 3x weekly)	[100,101]
sabarubicin	topoisomerase II inhibitor	human HCT116 (10–20 μM)	−	[116]
TRAIL	apoptosis inducer	human HT-29, human HCT116 (10 μM)	−	[91]
trichostatin A	histone deacetylase inhibitor	human HT-29 (10 μM)	−	[147]
Thoracic Cancers				
oxaliplatin	inter- and intra-strand DNA cross-linker	human A549 (10–500 μM)	−	[127]
paclitaxel (Taxol)	microtubules stabilizer	human A549, H446, human A549-T24, human H460 (5 μM)	human A549 and H460 in athymic nude mice (5 mg/kg 3x weekly)	[79,81,82]
valproic acid	histone deacetylase inhibitor nitric oxide synthase inhibitor	human TE2, human TE12, human H322, human H460, human H513, human H211 (20–30 μM)	−	[142]
Pancreatic Cancers				
actinomycin-D	RNA polymerase inhibitor	human PANC-1 (3–24 μM DMAPT)	−	[168]
ATO	reactive oxygen species inducer	human PANC-1, human BxPC-3 (2.5–5 μM)	human PANC-1 in athymic nude mice (0.8 mM/25 μL 2x weekly)	[158]
celecoxib	COX-2 inhibitor	−	cancer induced in hamster by N-nitroso bis(2-oxopropyl) amine (20–40 mg DMAPT/kg daily)	[97]
gemcitabine	ribonucleotide reductase inhibitor	human BxPC-3, human PANC-1, human MIA PaCa-2 (1–10 μM DMAPT)	−	[136]
sulindac	prostaglandin synthesis inhibitor	human BxPC-3, human PANC-1, human MIA PaCa-2 (2.5–10 μM)	−	[94]
Melanomas				
DTIC (dacarbazine)	DNA alkylator	human A-375, patient derived melanoma (3–24 μM)	−	[26]
doxorubicin	topoisomerase II inhibitor	human A-375, human 1205Lu, patient derived melanoma (10–24 μM)	−	[36]
Hepatomas				
5-fluorouracil	thymidylate synthase inhibitor	human BEL-7402 (5–100 μM)	−	[133]
fenretinide	reactive oxygen species inducer	human Hep 3B, human SK-HEP-1 (4 μM)	−	[154]
NS398	COX-2 inhibitor	human Hep 3B, human Hep G2, human PLC (5 μM)	−	[98]
Ro317549	protein kinase C-alpha inhibitor	human Choi-CK, human Cho-CK, human JCK1, human SCK (10–40 μM)	human Choi-CK and human SCK in BALB/cByJ-Hfh11^null^ nude mice (2.5 mg/kg daily)	[165]
llTRAIL	apoptosis inducer	human Hep 3B, human Hep G2, human SK-HEP-1 (15 μM)	−	[6]
Brain Malignancies				
5-aza-2′-deoxycytidine	DNA methyltransferase 1 inhibitor	human Daoy, human D283 Med (2–5 μM)	−	[129]
cisplatin	inter- and intra-strand DNA cross-linker	rat C6, human U-138 MG (10–25 μM), human U-118 MG (1.875–60 μM ACT001)	human U-118 MG in female nude BALB/c mice (200 mg ACT001/kg 6x each 7th day)	[115,128]
DHEA (dehydroepiandrosterone)	NF-κB signaling pathway inhibitor	murine AtT-20 (1–10 μM)	male BALB/c nude mice (200 μg/mouse daily)	[112]
doxorubicin	topoisomerase II inhibitor	rat C6, human U-138 MG (10–25 μM)	−	[115]
temozolomide	DNA alkylator	human LN-18, human T98G (10 μM)	human T98G in SCID mice (2.5–10 mg/kg daily)	[124]
Gastric Cancers				
cisplatin	inter- and intra-strand DNA cross-linker	human MKN-28, human MKN-45, human MKN-74/5-15 μM, human SGC-7901 (5–10 μM)	−	[27,34]
paclitaxel	microtubules stabilizer	human MKN-28, human MKN-45, human MKN-74 (3–9 μM)	human MKN-45 in female BALBc nu/nu mice (0.25–4 mg/kg daily)	[34]
Other Cancers				
ganciclovir Burkitt’s lymphoma	DNA polymerases inhibitor	human Raji (4–6 μM)	−	[28]
bicalutamide prostate cancer	androgen receptor blocker	−	human CWR22Rv1 in nude athymic mice (40 mg/kg daily)	[32]
Docetaxel prostate cancer	microtubules stabilizer	human CWR22Rv1 (0.5–10 μM)	human CWR22Rv1 in nude athymic mice (40 mg/kg daily)	[32]
geldanamycin ovarian carcinoma	Hsp90 inhibitor	human OVCAR-3, human SK-OV-3 (1–8 μM)	−	[164]
OKA (okadaic acid) retinoblastoma	phosphoserine/threonine protein phosphatase 1 and 2a inhibitor	human Y79 (0.25–0.5 μM)	−	[160]

4HT, 4-hydroxytamoxifen; AML, acute myeloid leukemia; ATO, arsenic trioxide; ATRA, all-trans retinoic acid; BALB/cByJ-Hfh11^null^, nude mice, Daejeon, South Korea; CAC, colitis-associated colon cancers; COX, cyclooxygenase; DHEA, dehydroepiandrosterone; DTIC, dacarbazine; FLT3, FMS-like tyrosine kinase 3; Hsp90, heat shock protein 90; mTOR, mammalian target of rapamycin; NF-κB, nuclear transcription factor-kappa B; NOD/SCID mice, nonobese diabetic/severe combined immunodeficient mice; NSCLC, non-small-cell lung cancer; OKA, okadaic acid; PI3K, phosphoinositide 3-kinases; SAHA, suberoylanilide hydroxamic acid; SCID, severe combined immunodeficiency; TRAIL, tumor necrosis factor-related apoptosis-inducing ligand.

**Table 2 pharmaceuticals-13-00194-t002:** Clinical evaluation of activity of PN and its derivatives.

	Phase	Purpose	Clinical Trial Registry
PN	I	pharmacokinetics and toxicity	none [18]
DMAPT	I	AML, ALL, and other blood-lymph tumors	none [unpublished] (United Kingdom)
ACT001	I/II	safety, tolerability, pharmacokinetics, recurrent glioblastoma	ACTRN12616000228482 (Australia & New Zealand) ChiCTR-OIC-17013604 (China)

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
