# Peer review of "Parthenolide as Cooperating Agent for Anti-Cancer Treatment of Various Malignancies"

_pharmaceuticals, 2020, doi:10.3390/ph13080194_

Round 1
Reviewer 1 Report
The manuscript by Sztiller-Sikorska and Czyz reviewed the roles of parthenolide as cooperating partners in the treatment of different malignancies. The chemotherapeutic potentials of parthenolide are gradually becoming an enticing option in combination chemotherapy. The authors did a commendable job in this review, summarizing several important findings. However, there are a few minor issues that the authors should address.
After the introduction section, the author should include a separate heading (Mechanism of action) to explain how parthenolide works. Given the nice organization and thorough coverage, it is a bit surprising that the authors did not mention the roles of parthenolide in inhibiting tubulin carboxypeptidase activity and DNMT1 in the mechanism of action. The authors did mention both these points sections where the authors discuss the combinatorial effects of parthenolide.
The authors should also mention the effects of parthenolide on cancer stem cells in a separate section or at least a paragraph. This field is gradually becoming very important.
Figure two is the central figure in this review. The authors should refer back to it in the sections where they discuss the roles of parthenolide in combination chemotherapy. In each case, the authors should mention whether the mechanism of action corresponds to sections 1, 2, 3, or 4 of figure 2.
Table 1 is comprehensive and very helpful. It would facilitate the reader more if the authors could include another column or two to enumerate if the studies were done in vitro or in vivo, and in each case mention what cell line/ models and doses were used.
There are a few typos here and there, for example, section 2.2 should read TRAIL as opposed to TRIAL.
Overall, this is a nicely organized manuscript, and a few minor changes should make it clearer to the readers.
Author Response
Answer to Reviewer 1
The manuscript by Sztiller-Sikorska and Czyz reviewed the roles of parthenolide as cooperating partners in the treatment of different malignancies. The chemotherapeutic potentials of parthenolide are gradually becoming an enticing option in combination chemotherapy. The authors did a commendable job in this review, summarizing several important findings. However, there are a few minor issues that the authors should address.
We thank you for reviewing our manuscript and your suggestions how to improve it. We addressed all the criticisms and revised the manuscript accordingly.
After the introduction section, the author should include a separate heading (Mechanism of action) to explain how parthenolide works. Given the nice organization and thorough coverage, it is a bit surprising that the authors did not mention the roles of parthenolide in inhibiting tubulin carboxypeptidase activity and DNMT1 in the mechanism of action. The authors did mention both these points sections where the authors discuss the combinatorial effects of parthenolide.
Parthenolide is a multitarget drug. In the review we focus on its activity in combination with anticancer drugs. We agree with the Reviewer that the role of parthenolide in inhibiting tubulin carboxypeptidase activity and DNMT1 should be underlined (see lines 77-80 and 80-81, respectively). As suggested we introduced a separate heading “Mechanisms of PN action” (line 42)
The authors should also mention the effects of parthenolide on cancer stem cells in a separate section or at least a paragraph. This field is gradually becoming very important.
As suggested we extended the information on effects of parthenolide on cancer stem-like cells in “Introduction” (lines 35-41) and mentioned them in “Conclusions and perspectives”.
Figure two is the central figure in this review. The authors should refer back to it in the sections where they discuss the roles of parthenolide in combination chemotherapy. In each case, the authors should mention whether the mechanism of action corresponds to sections 1, 2, 3, or 4 of figure 2.
We agree with the Reviewer that it is worth to refer to Figure 2 while discussing the role of parthenolide in combined treatment. This is introduced in the revised version of the manuscript.
Table 1 is comprehensive and very helpful. It would facilitate the reader more if the authors could include another column or two to enumerate if the studies were done in vitro or in vivo, and in each case mention what cell line/ models and doses were used.
We included two additional columns to Table 1 describing whether studies were done in vitro or in vivo. Names of cell lines and PN concentration were indicated for in vitro study and model and dose for in vivo study.
There are a few typos here and there, for example, section 2.2 should read TRAIL as opposed to TRIAL.
The language has been corrected.
Reviewer 2 Report
Journal Pharmaceuticals (ISSN 1424-8247)
Title: Parthenolide as cooperating partner for anti-cancer treatment of various malignancies
Authors: Malgorzata Sztiller-Sikorska , Malgorzata Czyz *
The review article by Sztiller-Sikorska and Czyz is intetesting during the period of researchers debating over the precision and transitional medicines. The review is comprehensive and gives most of up-to-date details regarding the Parthenolide and its derivatives. The references are up-to-date. However, the authors can make few changes before it comes for publication. A clear introduction with integrated discussion cum perspective section would be needed.
There are various potent agents like the one described here reach preclinical studies. However only few reach clinic. In this scenario, the author should describe how parthenoloid could be better than other agents. A separate table with current or previous clinical trial details is warranted.
The mechanism of action figure should be extended to show the clear mechanism based on previous studies used genetic approach. If the pathenoloid is acting thorough NFkB, how targeting is viable – the authors should discuss in their perspective.
The section two need to reorganized based on the the class of active compounds consider in combinational therapy. Hence, the authors may describe a common mechanism of action for each class. Also this will give an glimpse that the mechanism of action is same or different with different compounds.
Author Response
Answer to Reviewer 2
The review article by Sztiller-Sikorska and Czyz is intetesting during the period of researchers debating over the precision and transitional medicines. The review is comprehensive and gives most of up-to-date details regarding the Parthenolide and its derivatives. The references are up-to-date. However, the authors can make few changes before it comes for publication. A clear introduction with integrated discussion cum perspective section would be needed.
We thank the Reviewer for critical comments. We highly appreciate the positive opinion about the quality of our review article. We think that by addressing the Reviewer’s concerns on “Introduction” (lines 21-41) and “Conclusions and Perspectives” (lines 519-543), we improved the clarity and impact of our manuscript.
There are various potent agents like the one described here reach preclinical studies. However only few reach clinic. In this scenario, the author should describe how parthenoloid could be better than other agents. A separate table with current or previous clinical trial details is warranted.
While wide array of biological activity and low toxicity make parthenolide a very promising drug with multi-pharmacological potential, its bioavailability is unsatisfactory, which limits its clinical application in anticancer therapy. Therefore, parthenolide was used only in one clinical trial phase I to evaluate its pharmacokinetics and toxicity. Its derivative with better bioavailability, fumarate salt of dimethylamino-micheliolide (ACT001) is currently tested in China and Australia. Therefore, a Table 2, which was included in the revised version, is not especially rich.
The mechanism of action figure should be extended to show the clear mechanism based on previous studies used genetic approach. If the pathenoloid is acting thorough NFkB, how targeting is viable – the authors should discuss in their perspective.
The main aim of the review is to discuss a potential of parthenolide as a cooperating partner for diverse anticancer compounds, mainly clinically used drugs. This might be interesting in the perspective of better delivery system of PN. The mechanisms of action of parthenolide have been reviewed elsewhere.
The section two need to reorganized based on the the class of active compounds consider in combinational therapy. Hence, the authors may describe a common mechanism of action for each class. Also this will give an glimpse that the mechanism of action is same or different with different compounds.
In the revised version of the manuscript, we refer to Figure 2 while discussing the role of parthenolide in combined treatment. Mechanisms of action for each class of cooperating compounds have been introduced comprehensively in our opinion.
Round 2
Reviewer 2 Report
I am pleased with the authors reply and it address all my concerns.
Author Response
Thank you!